# Downregulation of Serotonergic System Components in an Experimentally Induced Cryptorchidism in Rabbits

**DOI:** 10.3390/ijms25063149

**Published:** 2024-03-09

**Authors:** Francisco Jiménez-Trejo, Cristian Arriaga-Canon, Luis A. Herrera, Isabel Coronado-Mares, Rogelio Montiel-Manríquez, Isaac González-Santoyo, Wendy B. Pérez-Báez, Miguel Tapia-Rodríguez

**Affiliations:** 1Laboratory of Cellular and Tissular Morphology, National Institute of Pediatrics, Insurgentes Sur No. 3700-C. Coyoacán, Mexico City 04530, Mexico; 2National Institute of Cancerology, Av. San Fernando No. 22, Tlalpan, Section XVI, Mexico City 14080, Mexico; cristiancanon@hotmail.com (C.A.-C.); rogelio_montiel@hotmail.com (R.M.-M.); phdwendybaez@ciencias.unam.mx (W.B.P.-B.); 3Biomedical Research Unit in Cancer, National Institute of Cancerology-Institute of Biomedical Research, UNAM, Tlalpan, Mexico City 14080, Mexico; herreram@biomedicas.unam.mx; 4Tecnológico de Monterrey, School of Medicine and Health Sciences, Monterrey 64710, Mexico; 5Tlanepantla Regional Hospital, Av. Paseo del Ferrocarril No. 88, Tlanepantla de Baz 54090, Mexico; isabelcoronado20@gmail.com; 6Faculty of Psychology, Universidad Nacional Autónoma de México, Circuito Escolar S/N, Ciudad Universitaria, Mexico City 04510, Mexico; isantoyo@psicologia.unam.mx; 7Microscopy Core Facility, Biomedical Research Institute, Universidad Nacional Autónoma de México, Tercer Circuito Exterior S/N, Ciudad Universitaria, Mexico City 04510, Mexico

**Keywords:** serotonin, testes, cryptorchidism, gonocytes, kynurenine

## Abstract

Cryptorchidism (CO) or undescended testes is defined as the failure of one or both testes to be positioned inside the scrotum. Typically, cryptorchidism is detected at birth or shortly thereafter, and in humans, it is considered to be part of the testicular dysgenesis syndrome (TDS), a complex pathology regarding the male reproductive system that apparently involves the interaction of both genetic and environmental harmful factors, mainly during embryonic development. Serotonin (5-HT) is an ancient molecule that participates in a broad range of body functions, and in recent years, its importance in reproduction has started to be elucidated. In male pathologies such as infertility, varicocele, erectile dysfunction, and primary carcinoid tumors, an increase in 5-HT concentration or its metabolites in the blood, semen, and urine has been directly related; nevertheless, the role of 5-HT in CO remains unknown. In the present work, our goal was to answer two important questions: (1) whether some serotonergic system components are present in adult male Oryctolagus cuniculus (chinchilla rabbit) and (2) if there are changes in their expression in an experimental model of CO. Using histological, molecular, and biochemical approaches, we found the presence of some serotonergic system components in the adult chinchilla rabbit, and we demonstrated that its expression is downregulated after CO was pharmacologically induced. Although we did not test the role of 5-HT in the etiology of CO, our results suggest that this indoleamine could be important for the regulation of steroidogenesis and spermatogenesis processes in the chinchilla rabbit during adulthood. Finally, in parallel experimental series, we found downregulation of kynurenine concentration in COI rabbits when compared to control ones, suggesting that CO could be affecting the kynurenine pathway and probably testicular immune privilege which in turn could lead to infertility/sterility conditions in this disorder.

## 1. Introduction

Cryptorchidism (CO) is a congenital disorder anatomically defined by the absence of one or both testes from the scrotal sac. This condition is included within testicular dysgenesis syndrome (TDS), which encompasses another three male urogenital pathologic conditions: infertility, hypospadias, and testicular cancer. CO is considered a disease of complex etiology in which many hormonal, genetic, physiological, and environmental factors are involved [1,2,3,4], and it is a public health problem because it (1) frequently results in infertility/sterility conditions and (2) increases the risk of developing testicular cancer, even after correcting the undescended testes with surgery treatments such as orchiopexy [5,6,7,8,9,10,11].

Several experimental studies performed in animal models have described a broad range of gonadal misfunctions that CO promotes: a reduction in testis size, the induction of smaller seminiferous tubule diameters, a delay in germ cell maturation, the total number of the reductions in and arrests of the differentiation of gonocytes, Sertoli cells’ tight junction damage, basal lamina thickening, an increase in apoptosis, disturbances in the local synthesis of hormones (androgens) and neurofactors (β2-adrenergic receptors), and severe decrement in both the quantity and quality of sperms [12,13,14]. In addition, the failure in the differentiation of gonocytes into spermatogonial stem cells could be responsible for the development of germ cell neoplasia in situ (GCNIS) and testicular germ cell tumor (TGCT) formation [3,4,5,15,16,17].

Normal testicular descent is highly dependent on an intact hypothalamic–pituitary–gonadal (HPG) axis [3,4,5]; if its balance is disrupted, it could result in CO. For this reason, it has been suggested that environmental or lifestyle (i.e., constant stress exposure, androgen insensitivity, premature babies, and low birth weight) rather than genetic factors are the most influential on the occurrence of this disorder [2,5,15]. It has been shown that exposure to environmental endocrine disrupting chemicals (EDCs) (oestrogenic and/or anti-androgenic compounds) during pregnancy will frequently result in genital abnormality and/or CO in the newborn male exposed. EDCs could affect differentiation in gonocyte, Sertoli, and Leydig cell populations which will result in a reduced testicular size; they also could disturb steroid hormone metabolism and modify the availability of neurofactors or neurohormones in testes affecting the HPG axis [5,15,16].

Although previously it was assumed that the neural control of testes was exclusively performed through the HPG axis, the description of alternative neural pathways, neuroendocrine cells, and the presence of neural markers in normal testes, which has been reported in recent years, have expanded our vision about the neural regulation of gonadal functions [15,17]. The presence of serotonin (5-hydroxytriptamine, C_10_H_12_N_2_O (5-HT)) has been described in mammal reproductive tissues, including testes [18,19]. The essential amino acid L-Tryptophan is the precursor of 5-HT synthesis and is regulated by the kynurenine pathway (KP), which is responsible for metabolizing most of the free tryptophan in mammals. The KP is activated by the induction and activity of key enzymes such as indoleamine-2,3-dioxygenase (IDO1), kynurenine-2,3-monooxygenase (KMO), and kynureninase. Pathological conditions and experimental models in which 5-HT or closely related molecules were pharmacologically modulated have suggested that 5-HT concentration must remain in normal ranges for a proper function of the testes [20,21,22,23,24,25,26]. Because of the apparent tight regulation of 5-HT concentration in gonads, we were interested in evaluating serotonergic system elements in an animal model in which testis homeostasis is disrupted; so, we decided to combine pharmacological, anatomical, biochemical, and molecular approaches to characterize the serotonergic system in chinchilla rabbits (*Oryctolagus cuniculus*), in which CO was pharmacologically induced (COI). Interestingly, we found downregulation of kynurenine and serotonin concentration in COI rabbits when compared to control ones; the biological effects of the various downstream metabolites of the kynurenine pathway have been linked with symptom development and disease progression in a wide range of disorders. The present results strengthen our proposal that this indoleamine could be important for the regulation of steroidogenesis and spermatogenesis processes during adulthood.

## 2. Results

### 2.1. Anatomical and Morphological Differences in Control and COI Testes

Table 1 shows morphometric parameters measured in control and COI rabbits; although there were no statistical differences in body weight between both groups, control rabbits showed higher GSI and TV values than the COI group (*p* < 0.05; n = 8, for each condition).

Figure 1 shows photographs of the inguinal region and exposed testes from both control and COI rabbits and representative images of their corresponding histological studies. Control rabbits 150 days old show a prominent pigmented scrotal sac and evident testes inside the scrotum and a relaxed penis (Figure 1A). After exposing them, abundant adipose tissue located in the upper part of the testes and thick blood vessels running through them are noted (Figure 1B; asterisks); also, the epididymis shows normal size and structure, according to the age of the animal. On the other hand, in COI rabbits, the scrotum and skin show no pigmentation or testis descent (Figure 1E), and the penis does not look relaxed. After exposing the region of the scrotum, a relatively abundant adipose tissue located in the upper part of the testes (Figure 1F; asterisks) was found. COI testes were of smaller size than control and showed blood vessels and epididymis according to their size. In histological sections of control rabbits, seminiferous tubules showed organized structure with the basal membrane, Sertoli cells, peritubular myoid cells, spermatogonia, spermatocytes, spermatids, and sperm all showing their respective characteristic morphologies, and Leydig cells and blood capillaries were found in the interstitial region with normal morphology (Figure 1C); it must be emphasized that at this stage gonocytes were absent. In sharp contrast, COI testes’ sections showed apparently normal blood capillaries and more numerous but rounder Leydig cells in the interstitial space; seminiferous tubules were found with smaller diameters and their cellular organization disrupted; peritubular myoid and Sertoli cells were apparently normal, but the basal membrane was thicker; gonocytes and shrunk spermatogonia but no spermatid were found inside of tubules (Figure 1G). Figure 1D shows the control rabbit TEM photomicrograph of a control seminiferous tubule in which type A spermatogonia are aligned (asterisks) and in close contact with the basal lamina with no presence of gonocytes, whereas in COI rabbit (Figure 1H), big round cells showing cytoplasmic low electrodensity, a prominent nucleolus, and few organelles, typical ultrastructural characteristics of gonocytes, were observed.

### 2.2. Determination of the Concentration Profiles of Both Serotonin and L-Kynurenine in Control and COI Testes

5-HT and L-Kynurenine concentrations were quantified in control and COI testis homogenates through ultra-sensitive ELISA kits (Table 2). Although L-Kynurenine was found more abundant than 5-HT, the control had higher concentrations of both molecules than COI testes (n = 6 for each condition).

### 2.3. Expression of Transcript Genes Related to the Serotonin System

Figure 2 shows a gene expression analysis that was performed through real-time PCR with the SYBR Green system for *Tph1*, *Maoa*, *Htr2a*, *Htr3a*, and *Slc6a4* of control and COI testes. We found that all but the *Slc6a4* transcript (5-HT_T_) show downregulation in COI rabbits when compared to control ones.

### 2.4. Distribution of Serotonergic System Elements in Control and COI Testes

Figure 3 shows representative photomicrographs of serotonergic system markers evaluated in the testicular slices of control rabbits; strong immunoreactivity for 5-HT was found in the interstitial zone on Leydig cells (star, Figure 3A) plus a weak immunoreactive signal in the nucleus; the inset shows a panoramic view. Relatively high TPH1 immunoreactivity (Figure 3B) was found in Leydig cells in the interstitial zone (star) and the neck and flagellum of spermatozoa (arrowhead, Appendix A); the inset shows positive neurons located in the brain stem of a rabbit used as positive control. 5-HT_1B_ receptor (Figure 3C and Appendix A) was found mainly expressed adjacent to the basal membrane of seminiferous tubules, in the cell membrane of Sertoli cells (arrowheads), in the perinuclear region of spermatogonials (arrow), and in the cytoplasm of Leydig cells (star). 5-HT_2A_ (Figure 3D and Appendix A) was found expressed in the cytoplasm of spermatogonial stem cells (arrowheads), adjacent to the basement membrane in peritubular myoids (blue arrow), in Sertoli-like cells (arrow), and secondary spermatocytes (asterisks); some sperm cytoplasmic droplets appeared stained in the lumen of some seminiferous tubules, plus a slight stain in the interstitial zone in Leydig cells (stars; the inset in Appendix A shows sperm immunoreactivity). The inset shows a panoramic view. The ionotropic receptor 5-HT_3A_ (Figure 3E and Appendix A) was observed strongly stained in the interstitial zone (star) and in the acrosomal region of several sperm (arrowheads and the inset show a panoramic view). MAO_A_ enzyme (Figure 3F) was found in Leydig cells (star), in spermatogonia (arrows), and presumptive spermatocytes in preleptotene (arrowheads), in the seminiferous tubule. The negative control slice is shown in the inset, with no observed signal expression.

Figure 4 shows photomicrographs of serotonin transporters immunostained in the testicular slices of control rabbits; 5-HT_T_ immunoreactivity was observed in the seminiferous tubules, presumably in spermatogonia and some Sertoli cells, in spermatocytes, spermatids, and mature spermatozoa, and in the interstitial zone in the walls of blood vessels (star), although the staining pattern was different for each cell type (Figure 4A and Appendix A). A higher magnification of a seminiferous tubule is shown in Figure 4B; spermatogonial stem cells show vesiculated diffuse cytoplasmic staining (arrowheads), Sertoli cells show diffuse staining (arrow), and both spermatocytes and spermatids show strong cytoplasmic expression (open long arrows), in the acrosomal region adjacent to the nucleus. Interestingly, spermatozoa also show strong immunoreactivity in the acrosome (arrows, Figure 4C); the inset shows a negative control. Figure 4D shows immunoreactivity against vesicular monoamine transporter (VMAT1) strongly stained in the interstitial zone (star) and expressed in the cytoplasm of Sertoli cells (arrowhead and Appendix A). The inset shows a panoramic view at lower magnification. L = lumen.

Some anatomical differences were found constant across all testicular slices obtained from COI rabbits and immunostained for serotonin markers when compared with control testes: seminiferous tubules with a smaller-diameter lumen, high circularity, reduced length, and a thicker basal membrane. 5-HT_1B_ receptor (Figure 5A and Appendix A) was found immunostained in the cell membrane of all cell layers of the seminiferous tubules (arrows) and within clustered Leydig cells in the interstitial zone (arrowheads); the inset shows positive neurons from the brain stem. 5-HT_2A_ receptor was found in cells adjacent to the basal membrane of seminiferous tubules, presumptively in spermatogonia-like cells and/or gonocytes (arrow), and in neighbor cells with cytoplasmic projections in the primary spermatocyte layer (arrowheads; Figure 5B and Appendix A); the inset shows a panoramic view of the distribution of this receptor. 5-HT3_A_ receptor was found distributed in the basal membrane in presumptive spermatogonia (arrows) and Sertoli cells (arrowheads) (Figure 5C and Appendix A); the inset shows a panoramic view of its distribution. MAO_A_ enzyme was found immunostained across all layers of the seminiferous tubules (Figure 5D); the inset shows negative control.

TPH1 enzyme was found intensely stained in the interstitial zone (arrowheads; Figure 6A) and slightly stained across seminiferous tubules (Appendix A); brain stem neurons are shown as positive control in Figure 6B (arrows), and the inset shows a negative control. 5-HT_T_ transporter was found intensely stained in clusters of Leydig cells (arrows) in the interstitial zone and in cell membranes of gonocytes (arrowheads), spermatogonials, and probably spermatocytes in the seminiferous tubules (Figure 6C and Appendix A); a higher-magnification image of a seminiferous tubule in which cell membranes of gonocytes (arrowheads) and spermatogonials (arrows) are intensely stained is shown in Figure 6D.

Double immunofluorescence for 5-HT and VMAT1 transporter is shown in Figure 7 of COI rabbits; 5-HT immunoreactivity was mainly distributed along the interstitial zone (stars; Figure 7A, single channel), whereas VMAT1 was distributed in both the interstitial zone and seminiferous tubules (Figure 7B, single channel, and Appendix A), apparently in the Leydig (stars) and Sertoli cell cytoplasm, respectively. Figure 7C shows an apparent colocalization in some Leydig cells but not in others; apparently, there are three Leydig cell populations present: one 5-HT only positive, another VMAT1 only positive, and a small population 5-HT/VMAT1 double positive. Brain stem neurons are shown as positive control for 5-HT (arrows; Figure 7D); the inset shows a negative control. L = lumen.

### 2.5. Downregulation of Proteins Measured with Immunoblotting

Western blot (Wb) analyses were carried out for TPH_1_ (~48 kDa), MAO_A_ (~61 kDa), 5-HT1_B_ (~47 kDa), 5-HT_2A_ (~53 kDa), 5-HT_3A_ (~48 kDa), and VMAT1 (~55 kDa) for both control and COI testes. Figure 8A shows the representative Western blots of all serotonin system proteins evaluated in both control and COI rabbits’ homogenates. As noted, there was an important reduction in COI testes in all markers evaluated when compared to control ones. Densitometric analyses of protein bands were performed, and they confirmed both qualitative and semi-quantitative differences between control and COI rabbits (Figure 8B).

## 3. Discussion

In mammals, the physiology of the testes during sexual maturity is a complex process of multiple well-controlled hormonal, cellular, and molecular mechanisms. Although its main control is wielded by the hypothalamic–pituitary–gonadal axis (HPG axis), the fine regulation of key processes such as steroidogenesis and spermatogenesis relies on molecules that belong to other systems [27,28]. For this reason, it has been suggested that 5-HT could regulate, directly or indirectly, some male reproductive functions including steroidogenesis [29,30], spermatogenesis [31,32,33], sexual maturity in mammals [34,35,36], and urogenital pathologies in humans [37,38], including testis cancer [39,40,41] and infertility [18,19].

Congenital cryptorchidism, a failure of one or both testes to descend into the scrotal sac, is one of the strongest risk factors for developing testicular cancer, testicular torsion, and/or an infertility/sterility condition [2,13,42]. In humans, although its etiology is multifactorial and remains largely uncertain (for a review, see [43,44]), there are some risk factors well established for it, such as prematurity (15–30%), family history and genetic mutations, low birth weight for gestational age, estrogen exposure during pregnancy, and smoking habit, among others. Also, some of the cellular and physiological consequences that CO produces on testes have been well documented [5,13,14,15,16,45]. Because of the broad anatomical and physiological changes that occur in CO testes, we were interested in analyzing if serotonergic system elements were affected in this disorder. To address this issue, we used 17 α-estradiol (E2) as an endocrine-disrupting chemical (EDC) to induce CO in rabbits, as has been described before [46]. In the chinchilla rabbit breed, it has been reported that testes’ descent is fulfilled around 50 days postpartum and that constant inoculation of E2 during the postnatal period induces CO in them [45,46]. In agreement with those previous studies, we found evident anatomical differences between our age-matched control and COI rabbits. E2 treatment induced significant decrements in both GI and TV in COI rabbits when compared to control ones. Also, the cellular organization of testes was severely altered in COI rabbits when compared to control ones. Although the mechanisms that lie behind these responses remain controversial, it has been suggested that E2 acts on estrogen receptor alpha in Leydig cells, inhibiting the expression of insulin-like factor 3 (INSL3), a secretory protein that has been closely related to the transabdominal phase of testicular descent [47,48,49]. The smaller size of seminiferous tubules, the notorious presence of gonocytes, and the absence of spermatids suggest that both steroidogenesis and spermatogenesis processes are disrupted in COI rabbits at 150 days of age. Interestingly, in these animals, gonocytes migrated toward the basal membrane instead of remaining in the center of the seminiferous tubules, which are characteristic of neonatal gonocytes [7,50]. In addition, the morphology of spermatogonia-like cells appeared irregular and shrunk, while spermatids were absent when compared with control rabbits.

Besides the important anatomical differences previously described between COI and control rabbits, we explored some elements of the serotonergic system in both groups of rabbits. The enzyme responsible for 5-HT synthesis outside the CNS, tryptophan hydroxylase 1, was found expressed in Leydig cells in both control and COI rabbits, in the sperm of control rabbits, and slightly stained in the seminiferous tubules of COI rabbits. The expression of this anabolic enzyme is maintained in the interstitial space of COI rabbits, suggesting that local 5-HT synthesis by Leydig cells is potentially not affected by the CO condition. In sharp contrast, the presence of immature gonocytes and the absence of sperm in COI rabbits are consistent with a failure in spermatogenesis. It has been previously reported that TPH1 is expressed in mature sperm, at least in horses, rats, and humans [32,51,52]. On the other hand, the catabolic monoamine oxidase A (MAO_A_) enzyme was found in the Leydig cells, spermatogonia, and spermatocytes of control rabbits and across all cell layers of seminiferous tubules but not in Leydig cells in COI rabbits. MAO_A_ enzyme degrades 5-HT, dopamine, norepinephrine, and tyramine through oxidative deamination, and its activity has been related to the modulation of 5-HT concentration when the anabolic TPH enzyme is present too [30,51,53]. If Leydig cells from COI rabbits lack this enzyme, it is probably that an imbalance in the concentration of this indoleamine could affect steroidogenesis, but this condition could be “rescued” by another serotonergic element, such as 5HT_T_ (see below). In the same way, control sperm does not show MAO_A_ expression as it has been described in horses, rats, and humans [32,51,52], so there is a probability that 5HT_T_ is regulating the concentration in the sperm head too. 

5-HT can exert diverse, sometimes opposite, physiological actions, and these responses are determined by the receptor subtypes it interacts with and the intracellular signaling pathways coupled to them [54,55,56]. In the present work, the gene expression, presence, and localization of 5-HT_1B_, 5-HT_2A_, and 5-HT_3A_ receptors were evaluated, and noticeable changes in the distribution of serotonin receptor subtypes evaluated were found. In control rabbits, 5-HT_1B_ receptor was expressed in presumptive peritubular myoid and Sertoli cells, spermatocytes, and Leydig cells whilst in COI rabbits, its signal was only found in spermatocyte-like and Leydig cells. In control rabbits, 5-HT_2A_ receptor was found immunostained in spermatogonia, the peritubular myoid, Sertoli-like cells, secondary spermatocytes, and spermatozoa, whereas in COI rabbits, this receptor was found expressed also in presumptive gonocytes. In peripheral tissues, the 5-HT_2A_ receptor is expressed in vascular smooth muscle cells mediating vasoconstriction whilst the 5-HT_1B_ receptor has been linked to endothelial cells mediating vasodilation [57,58]. Interestingly, both receptors seem to be expressed in the peritubular myoid cells of control rabbit testes, which are specialized smooth muscle cells. The lack of expression of 5-HT_1B_ receptors in the peritubular myoid cells of COI rabbits could be a disrupting factor for the failure in spermatogenesis because peritubular myoid cells are important for spermatogonial stem cell differentiation and for intratesticular cell transport from the basal membrane to the lumen [59,60]. It must be noted that both receptors are also expressed in the Sertoli cells of control rabbit testes, but the 5-HT_1B_ expression in these cells is severely diminished in COI rabbits, probably contributing to the alteration of spermatogenesis in these animals. On the other hand, the ionotropic 5-HT_3A_ receptor was found in the interstitial zone and the sperm acrosome of control rabbits, whilst in COI rabbits, it was found expressed in presumptive gonocytes, spermatogonia, and Sertoli cells. Subunit A of this receptor is capable of homopentamerizing and triggering fast ionic responses of monovalent and divalent cations across the cell membrane in response to 5-HT in these cell types, as it occurs in the CNS, brain–gut circuitry, and other neural cells [54,56,61]. 

The presence of 5-HT3_A_ in the sperm acrosome in control rabbits is in agreement with the finding of these receptors in other mammal species including humans [32,51,52] and reinforces the hypothesis that these cation-selective 5-HT-gated ion channels could be important for the sperm movement when they are in the female tract. The presence of these receptors in the interstitial zone was unexpected, and in a previous work of the description of indoleaminergic elements in normal rat testes 5-HT3_A_, it was found in the seminiferous tubules but not interstitial space [51], but in an earlier work, Dufau et al. (1993) found that 5-HT could modulate the corticotropin-releasing factor (CRF) through 5HT_2_ receptors in Leydig cells [30]. Although 5HT3 agonists were not able to induce the release of CRF, it is possible that 5-HT3_A_ receptors could participate in other cellular processes, but further experiments are required to test this statement. Same as in normal rat testes, the 5-HT3_A_ receptor was found in the seminiferous tubules of COI rabbits, although in these animals, the presence of gonocytes is an abnormal condition per se and could be a key factor for the presence of an infertility/sterility condition in these animals. Although we searched for three 5-HT receptors, we are aware that there are other eleven subtypes that could be present in testes, and the complexity of responses to 5-HT could be greater than our proposal [56]. 

We evaluated 5-HT transporters, and there were no evident differences in their distribution. In control rabbits, 5-HT_T_ immunoreactivity was observed in spermatogonia, some Sertoli cells, spermatocytes, spermatids, mature spermatozoa inside seminiferous tubules, and the walls of blood vessels in the interstitium, and VMAT1 immunoreactivity was found in the interstitial zone and Sertoli cells. In COI rabbits, 5-HT_T_ transporter was found in Leydig cells in the interstitial zone and cell membranes of gonocytes, spermatogonia, and spermatocytes in the seminiferous tubules while VMAT1 was found in both Leydig and Sertoli cells. 5-HT_T_ is a Na^+^/Cl^−^-dependent transporter that is found oligomerized in the cell membrane and supports inward serotonin transport to the cytoplasm in normal conditions; however, in the presence of some stimuli or when there is a reversal of the Na^+^ gradient, it performs outward serotonin transport extruding this indolamine to the extracellular matrix [58,62]. Because of the lack of MAO_A_ enzyme in the Leydig cells of COI rabbits and the sperm head of control ones, there is a good possibility that 5HT_T_ could modulate the 5-HT concentration in these cell types, but functional experiments must be performed to test this idea. In another way, VMAT1 is a transporter typically found in neuroendocrine cells that removes 5-HT from the cytoplasm and stores it in secretory vesicles with the aid of 2 H^+^ as antiporters [63,64,65]. In the present work, we found no differences in the distribution of VMAT1 between control and COI rabbits, although the transcript levels of this transporter were the only ones not affected by CO, and protein levels were down to about fifty percent in COI rabbits, which seems enough for a relatively proper function in this condition.

Finally, we quantified 5-HT concentration levels through ELISA assays, and we found that they were kept relatively high when related to the decreased levels of TPH1 enzyme present in COI testes. This could be attributable to the “peripheral” affluence of 5-HT from several sources like the bloodstream, mast cells located in the testicular capsule [66,67], nerve fibers that run through the interstitial zone [51], and importantly, the presence of VMAT1 in the Leydig cells of COI rabbits. Although 5-HT has been linked to testicular pathologies such as varicocele [37], which induces an increase in reactive oxygen species (ROS), it seems unlikely that this indoleamine could be responsible for the main morphological changes that are present in COI testes; instead, the disruption of the local serotonergic system could be affecting late phases of spermatogenesis both directly and indirectly [31,33]. In addition to 5-HT, we exclusively evaluated L-kynurenine concentration in control and COI rabbit testes through ELISA assays. L-Kynurenine (kyn) is produced by the enzyme indoleamine-2,3-dioxygenase (IDO) and belongs to the main metabolic pathway of the essential amino acid tryptophan, which also participates in the synthesis of proteins and serotonin [53,68,69]. Interestingly, the kynurenine pathway (KP) has been related to immune privilege in the eye, brain, placenta, epididymis, and testis [70]. Furthermore, it has been reported that IDO is expressed in the principal cells of the epididymis, and it is considered that Sertoli cells can maintain this balance of synthesis, protection, and a similar function in the testis as principal cells do in the epididymis [68]. In our experiments, we found downregulation of kyn in COI animals when compared to control ones, suggesting that CO could be affecting the KP and probably testicular immune privilege which could promote infertility/sterility conditions.

## 4. Materials and Methods

### 4.1. Animals and E2 Treatment to Induce Cryptorchidism

Animal handling was performed under a strict agreement of the guidelines established by the ethics committee of the Instituto Nacional de Pediatría (INP; 034-2015 approval). Chinchilla rabbits were bred and housed in the animal care facility of the INP in accordance with the Mexican Official Norm NOM-062-ZOO-1999 (Technical Specifications for the Production, Care and Use of Laboratory Animals. D.O.F. 22-VIII-2001). They were kept in special cages with water and food ad libitum through pharmacological treatment until they were euthanized.

Two groups (n = 12 each, control and COI) of rabbits (*Oryctolagus cuniculus*, European Chinchilla breed) were randomly assorted, and COI rabbits were treated as follows: seven days after birth, periodic subcutaneous administration of 17β-estradiol 3-benzoate (E2, acting as an endocrine disruptor (ED), SIGMA, Burlington, MA, USA) was started, inoculating 16.6 µg E2 every third day (as a diluent, corn oil was used, total dose of 500 μg); this pharmacological protocol acts on a temporal window in which testis descending is irreversibly prevented [46,71]. The control group was inoculated with a vehicle solution. It is known that the chinchilla rabbit reaches sexual maturity at 4.5 months of age, so all individuals were euthanized and castrated at 150 days old when they had fulfilled sexual maturity (see below).

### 4.2. Dissection, Collection and Preservation of Testes

For animal euthanasia, an initial dose of xylazine hydrochloride (2 mg/kg bw, Bayer Laboratories), combined with Tiletamine hydrochloride and Zolazepam (10 mg/kg bw, Zelasol, Zoetis Laboratory), was used for anesthetic effect. When the animals were anesthetized, they were carefully shaved and disinfected in the scrotum and neighboring regions, and a longitudinal incision was made in the midline of the scrotal sac to expose the tissue layers surrounding the testes, including the Lamina Parietalis of Tunica Vaginalis. Then, the testicular cords were located, ligated, and cut to release the testes with epididymis. Concluding the dissection, the scrotal sac was sutured. Immediately after the testes were dissected, they were sectioned to obtain tissue portions for the different techniques we employed. Tissue fragments used for immunofluorescence, Western blot, ELISA, and qRT-PCR were frozen by immersion in hexane pre-chilled with dry ice and stored at −70 °C until use [51]. Tissue portions used for electron microscopy analysis were embedded in Karnovsky fixative.

### 4.3. Gonadosomatic Index (GSI) and Testes’ Volume (TV)

The body weight of rabbits was measured with a standard balance, and after dissection, isolated testes were weighed with an analytical balance; with those data, a gonadosomatic index (GSI) was obtained [72,73]. GSI is the ratio of gonad mass in relation to the total body mass, and it is obtained by the formula GSI = [Gonad Weight/Total Body Weight] × 100 [72]; we used GSI for a correlative measure of sexual maturity with the testes’ development. In addition, we measured three different diameters using Vernier’s scale to obtain the testes’ volume (TV, [27]): anteroposterior (D1), dorso-ventral (D2), and mid-lateral (D3). From D1, D2, and D3, their respective radii r1, r2, and r3 were calculated (D1/2, D2/2, and D3/2). Once they were calculated, the following formula was used to calculate TV: TV = (4/3)π(r1)(r2)(r3) (n = 12 each, control and COI).

### 4.4. Quantitative Determination of Serotonin and L-Kynurenine in Control and COI Testes

Serotonin concentration in rabbit testis fragments of both control and COI groups was evaluated using the Enzyme Linked Immunosorbent Assay (ELISA colorimetric detection). Mechanical extraction was manually performed with disposable pistils using a cold Glycine-HCl buffer (0.2 M), with a pH of 2.2 (1:20 *w*/*v*). Immediately, four cycles of 30 s of sonication followed by 30 s in ice were performed with a sonicator (ultrasonic cleaner VGT-800; SharperTek^TM^, Pontiac, MI 48341, USA). Subsequently, the homogenates were centrifuged at 14,500 RPM for 40 min at 4 °C. The supernatants obtained were stored at −70 °C until use [74]. We used an Ultra-sensitive ELISA kit according to the protocol suggested by the manufacturer for serotonin determination (ENZO, ADI-900-175 Farmingdale, NY, USA) and the commercial sensitive L-Kynurenine ELISA kit (ImmuSmol, BA E-2200, Bordeaux, France) for detecting L-Kynurenine in testis samples using Absorbance Microplate Reader (BioTek Epoch Microplate Spectrophotometer, Winooski, VT, USA; triplicates; n = 6, for each condition). Results were expressed in nanograms/mL for 5-HT or µg/mL of homogenized tissue samples for L-Kynurenine.

### 4.5. RNA Extraction and RT-PCR

Total RNA from rabbit testis fragments were isolated according to Peña-Llopis and Brugarolas [75]; RNA integrity was verified using TapeStation 2100 bioanalyzer (Agilent, Santa Clara, CA, USA) following the manufacturer’s instructions, and all samples had RIN values higher than 8. Each sample was treated with DNase I, RNase-free (Thermo Scientific, Waltham, MA, USA, Cat. EN0521, according to manufacturer’s instructions), and they were reverse-transcribed using MuLV reverse transcriptase from Thermo Scientific (GeneAmp^®^ RNA PCR Core Kit, cat. N8080143) following the instructions of the manufacturer, except for the enzyme for which the final concentration in each reaction was 1.9 U/µL. We used 375 ng of total RNA in reactions of 20 µL. Then, PCR was performed using the following primers: 

***Tph1*** forward 5′-GACCACCCTGGCTTCAAAGA-3′

***Tph1*** reverse 5′-GCAAGCATGGGTCGGATAGA-3′

***Maoa*** forward 5′-ACCGAAACCGGGAGTTCATC-3′

***Maoa*** reverse 5′-CCGCCACTCAGACTGGATAC-3′

***Htr2a*** forward 5′-ACAGGGAGGGAGGATCTGAC-3′

***Htr2a*** reverse 5′-CGGGTTGAGCTTTCTCCAGT-3

***Htr3a*** forward 5′-GAACTGCAGCCTGACCTTCA-3′

***Htr3a*** reverse 5′-TGCAGGATGCCGTACATTGA-3′

*Slc6a4* forward 5′-GGTACATGGCGGAGATGAGG-3′

*Slc6a4* reverse 5′-CTGTCCAAGCCCAGTGTGAT-3′

Reference gene **HPRT** (hypoxanthine phosphoribosyltransferase 1) Oc03399461_m1 [76].

### 4.6. Quantitative RT-PCR

Quantitative RT-PCR measurements (qRT-PCR) were performed with QuantStudio3 (Applied Biosystems, Waltham, MA, USA); for nucleic acid staining, we used Maxima SYBR Green/ROX qPCR Master Mix (2X) (ThermoFisher, cat. K0221, Waltham, MA, USA). The PCR cycle used for the measurements was 50 °C for 2 min, 95 °C for 10 min, and 40 cycles of 95 °C for 15 s and 60 °C for 30 s with fluorescence capture at this step, followed by the melt curve stage of 95 °C for 15 s and 60 °C for 1 min, followed by temperature increase with ramp 0.5 °C/s, reaching 95 °C for 1 s, during which the fluorescence measurements were performed. The ramp for the rest of the protocol was 1.6 °C/s. Quadruplicates were made for each sample measurement. Relative mRNA levels of all genes (*Tph1*, *Maoa*, *Htr2a*, *5htr3*, *Slc6a4*) were calculated using the ΔΔCt method [75,76]. The transcript levels were normalized with HPRT expression (n = 4 for each gene analyzed).

### 4.7. Transmission Electron Microscopy (TEM) Analysis

Organs were processed for microscopy according to the techniques described previously [77]. Briefly, testes’ tissue portions embedded in Karnovsky solution were post-fixed in osmium tetroxide solution (OsO_4_; Merck, Darmstadt, Germany) and processed for inclusion in Epon 812 (Ted Pella, Inc., Redding, CA, USA). Thin and semithin slices of 70 nm and 1 µm thickness, respectively, were cut using an Ultracut UCT microtome (Leica Microsystems, Wetzlar, Germany), then were stained with uranyl acetate and lead citrate, and examined under a JEM-1011 (JEOL Ltd., Tokyo, Japan) transmission electron microscope (high contrast 2 k × 2 k AMT mid-mount digital camera). Spermatogenesis stages and the presence of gonocytes at adulthood were analyzed in the acquired photomicrographs of control and COI rabbits.

### 4.8. Immunofluorescence

Testicular sections (8–12 μm) were obtained on a cryostat (T = −20 °C) and mounted on gelatin-coated slides [74]. Sections of control and COI groups plus a brain stem section—used as a positive control of markers of the 5-HT system—were mounted on a slide for each marker evaluated. Immediately after cutting, sections were fixed by immersion in methanol-acetone (1:1) for 10 min and then washed with phosphate buffer saline (PBS; 0.1 mM, pH 7.4) at room temperature. Subsequently, antigen retrieval was performed with an Immuno/DNA retriever with citrate buffer (Bio SB; Santa Barbara, CA, USA) for 20 min at 60 °C. After washing, samples were incubated with blocking serum for 2 h (5% bovine serum albumin in PBS at pH 7.4). Then, primary antibodies were added overnight, diluted in 1:100 blocking serum. Information about the specificity and cross-reactivity of each antibody is provided in Table 1. After 3 washes with PBS, sections were incubated for two hours at room temperature with the corresponding secondary antibodies (Alexa Fluor 488 donkey anti-goat IgG and Alexa Fluor^TM^ 488 Goat anti-Mouse IgG (H + L) diluted 1:800–1:1200 in blocking solution). After washing, slides were embedded with fluorescence mounting medium (DAKO, Agilent Technologies, Inc., CA, USA) and coverslipped. In control experiments, slides were incubated with preimmune serum (TPH1 and 5-HT_2A_), or in all others, the primary antibodies were omitted. Experiments were performed in triplicate. Stained sections were visualized, and representative images were acquired using an epifluorescence microscope (Olympus BX-51, Olympus Corporation, Tokyo, Japan), a CCD digital camera (Olympus DP25, with software DP2-BSW, Olympus) controlled by Image-Pro Plus 7.0 (Media Cybernetics, Inc., ML, USA) software. Figure panels were elaborated using Adobe Photoshop software 10.0.1 or Adobe Photoshop CS5.1 version 12.1 X64 (Adobe Systems Incorporated, San Jose, CA, USA).

### 4.9. Immunotransference by Western Blot

Testes’ fragments of both control and COI were mechanically homogenized (Polytron) in cold lysis buffer (100 mM NaCl, 1 mM PMSF, 1% Triton-X100 with a protease inhibitor cocktail tablet, Complete™ Roche, dissolved in 0.01 M Tris-HCl pH 8.0). Tissue homogenates were centrifuged at 14,500 RPM at 4 °C for 30 min; the supernatant was collected, and total protein determination was performed with a NanoDrop^TM^ Lite microvolume spectrophotometer (ThermoFisher Scientific). Gels were made with 12% sodium-polyacrylamide dodecyl sulfate 0.75 mm thick based on the protocol described by Jiménez-Trejo et al. [74]. We used the same antibodies from Table 1 to confirm the presence of serotonin system markers in testis homogenates (n = 4 for each experiment).

### 4.10. Statistical Analysis

Value data were expressed as mean ± S.E.M. Statistical differences between groups were assessed by two-way analysis of variance (ANOVA) for the data shown in Table 1 and Table 2 and Figure 2 and Figure 8B and by Student’s t-test for the remaining data. All statistical analyses were performed using Statistical Analysis Software (SPSS, version 12.0; SPSS Japan, Inc., Shibuya-Ku, Japan). *p* < 0.05 was considered significant.

## 5. Conclusions

In the present work, we found that there are some serotonergic system elements present in chinchilla rabbit testes, and we suggest that in COI rabbits, an alteration of some of them occurs. We support our immunohistochemical results with real-time PCR and Western blot techniques where we found downregulation of both gene expression and protein concentration in COI rabbits. Altogether, these experiments strengthen the idea of the presence of a conserved, local serotonergic system in the reproductive system of eutherians and strongly suggest that in cryptorchidism, there is a disruption of it, with functional consequences. The presence of some serotonin receptors and transporters that has been previously described in the reproductive system of another species suggests that the serotonergic system could be acting in chinchilla rabbit testes in a similar way as we and others have proposed previously, contributing to the regulation of both steroidogenesis and spermatogenesis processes in a tight, very controlled system [5,25,26,47]. The disruption of the 5HT system in COI rabbits probably does not determine the anatomical and physiological changes that were observed, but it could negatively affect the spermatogenesis process (see Figure 9), although further experiments must be performed to test this idea. Finally, the use of serotoninomics [22,78,79], meaning the evaluation of all serotonergic elements at a time, in future studies in humans and animals will give more insights into the role of the serotonergic system in complex reproductive pathologies.

## Figures and Tables

**Figure 1 ijms-25-03149-f001:**
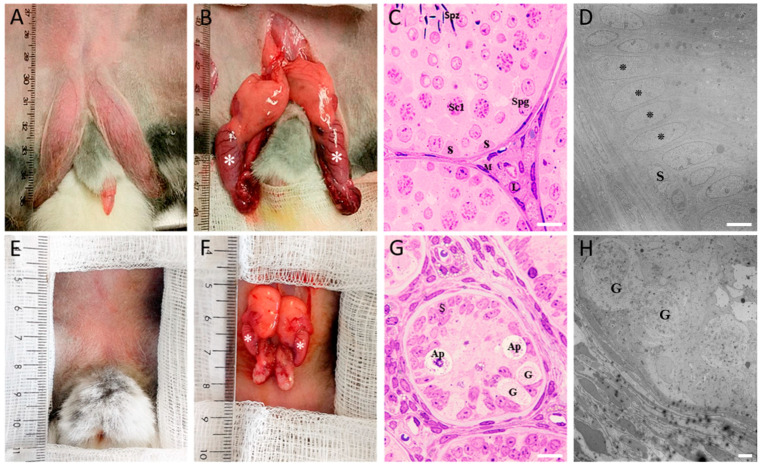
Anatomic description of control and COI rabbits. Representative images of intact testes and dissected in control ((**A**,**B**) respectively) and COI (**E**,**F**) rabbits; asterisks in (**B**) and (**F**) denote the testis; absence of both testicular descent and a pigmented scrotal sac is noted in COI rabbits when compared to control ones. (**C**,**G**) show representative photomicrographs of testis sections stained with H&E, in control (**C**) and COI testis (**G**). Representative TEM microphotograph of 150-day-old normal (**D**) and COI (**H**). Please see details in text. (**H**) S = Sertoli cell; L = Leydig cell; M = peritubular myoid cell; Spg = spermatogonium; Sc1 = primary spermatocyte; Spz = sperm; Ap: atypical cells. G = gonocyte. Scale bar in (**C**,**G**): 10 µm, (**D**): 5 µm, and (**H**): 7 µm.

**Figure 2 ijms-25-03149-f002:**
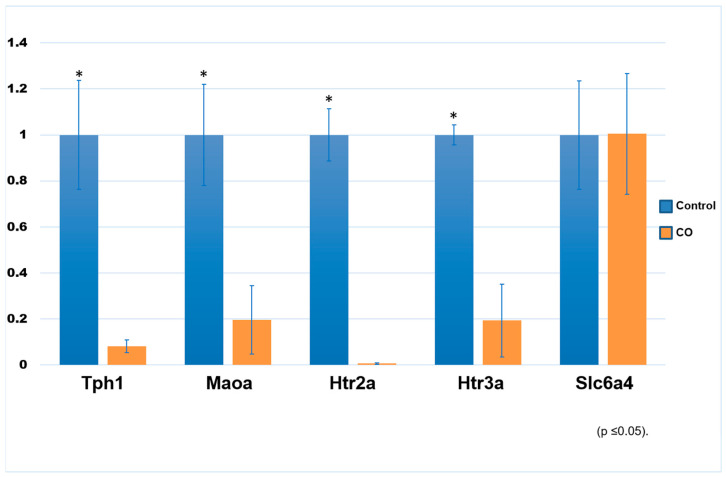
Gene expression of some serotonergic system elements in control and COI testes. Relative mRNA levels of *Tph1*, *Maoa*, *Htr2a*, *Htr3a*, and *Slc6a4* (5HT_T_), in both control and COI rabbits, are shown. * denotes statistical significance differences between experimental conditions. *p* value was set at ≤0.05.

**Figure 3 ijms-25-03149-f003:**
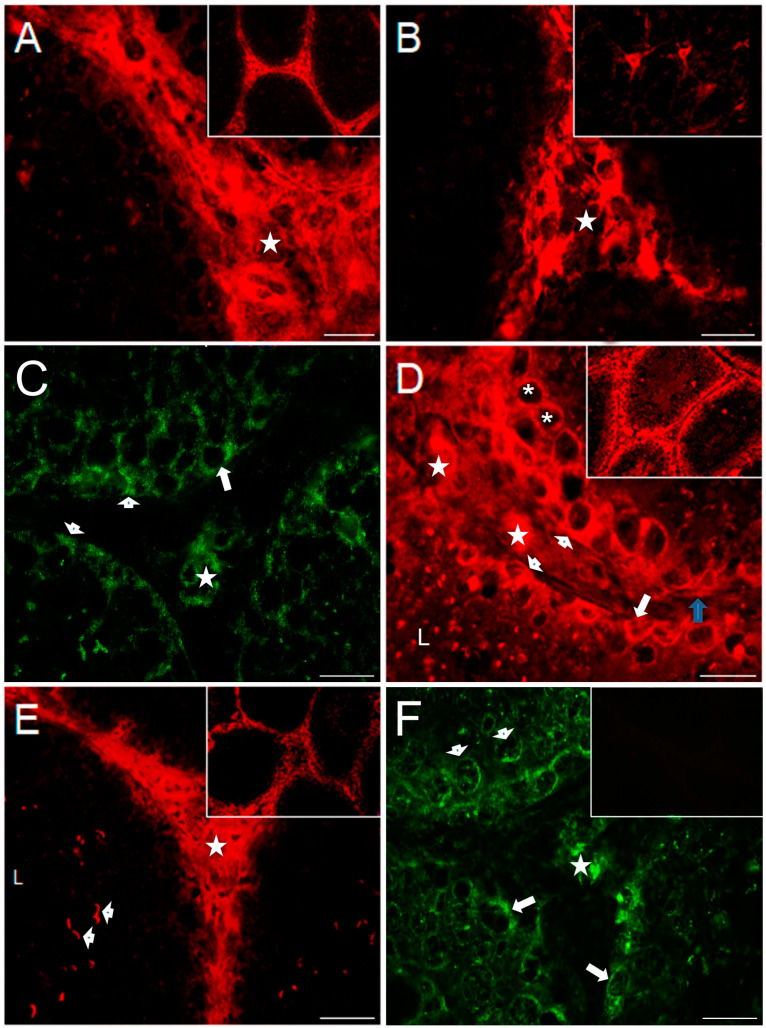
Immunofluorescence of serotonergic system elements in 150-day-old control rabbit testes’ sections (I). 5-HT was found strongly immunostained in interstitial zone (asterisks), and a weaker signal for it was found in the seminiferous epithelium (**A**); inset is a panoramic view. TPH1 was found strongly immunostained in clusters of Leydig cells in the interstitium (asterisks); inset shows brain stem neurons used as positive control of technique (**B**). 5-HT_1B_ receptor (**C**) was found immunostained in Leydig cells (star), Sertoli-like cells (arrowheads), and the perinuclear region of spermatogonials (arrow). 5-HT_2A_ receptor (**D**) was found immunostained in spermatogonials (arrowheads), Sertoli-like cells (arrow), secondary spermatocytes (asterisks), and some sperm cytoplasmic droplets in the lumen and was slightly detected in the Leydig cell in interstitium (stars). 5-HT_3A_ receptor (**E**) was found strongly immunostained in the interstitial zone (star) and in the sperm acrosome (arrowheads); inset shows a panoramic view. MAO_A_ enzyme (**F**) immunoreactivity was found in Leydig cells (star), spermatogonials (arrows), and putative spermatocytes in preleptotene (arrowheads); inset shows a negative control. Scale bar in (**A**–**F**): 20 µm.

**Figure 4 ijms-25-03149-f004:**
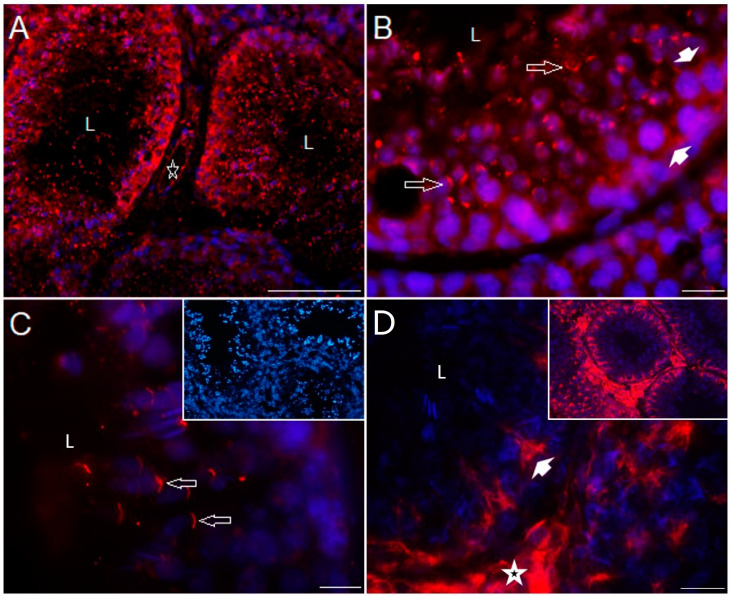
Immunofluorescence of serotonergic system elements in 150-day-old control rabbit testes’ sections (II). Panoramic view of 5-HT_T_ immunolocalization in the seminiferous tubules, presumptively in Sertoli cells, spermatogonia, spermatocytes, spermatids, and sperm, and in the interstitial zone in the walls of blood vessels ((**A**); star); higher magnification of a seminiferous tubule (**B**), in which spermatogonial stem cells show diffuse cytoplasmic staining (arrowheads), Sertoli cells show more vesiculated, diffuse staining (arrow), and both spermatocytes and spermatids show strong cytoplasmic expression (open arrows), presumably in the acrosomal region adjacent to the nucleus; in the same way, spermatozoa (higher magnification at (**C**)) show strong immunoreactivity in the acrosome (open arrows), and inset shows a negative control; VMAT1 (**D**) was found strongly stained in the interstitial zone (star) of seminiferous tubules, probably in Sertoli cells (arrowhead), and inset shows a panoramic view at lower magnification. L: lumen of seminiferous tubules. Scale bar in (**A**): 100 µm and (**B**–**D**): 10 µm.

**Figure 5 ijms-25-03149-f005:**
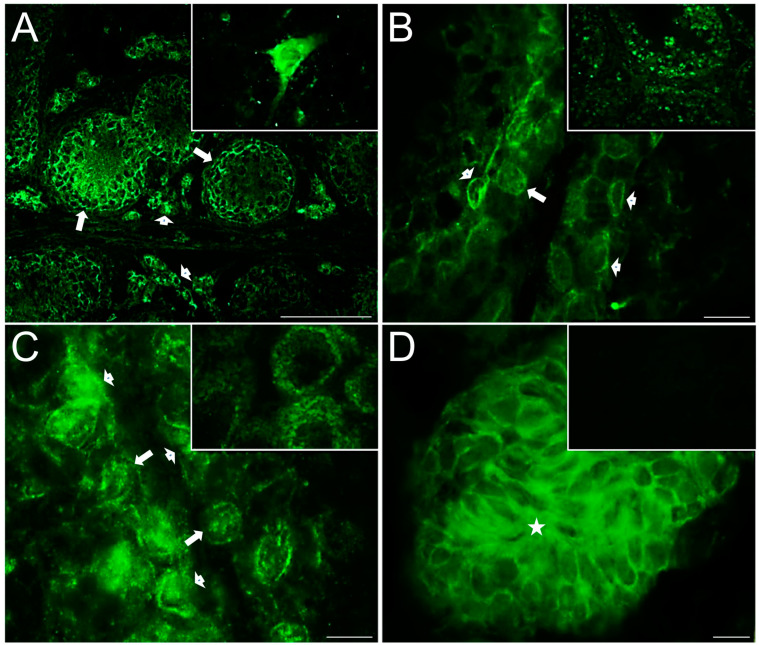
Immunofluorescence of serotonergic system elements in 150-day-old COI rabbit testes’ sections (I). Panoramic view of 5-HT_1B_ receptor immunolocalization (**A**), which was found immunostained all through the small seminiferous tubules in the cell membranes of apparently all cell types belonging to them (arrows) and in clusters of Leydig cells (arrowheads); inset shows a neuron from brain stem used as positive control. 5-HT_2A_ receptor immunoreactivity (**B**) was found in Sertoli-like cells (arrowheads) and gonocytes (arrow); inset shows a panoramic view at lower magnification. 5-HT_3A_ receptor (**C**) was found expressed in presumptive gonocytes and Sertoli cells (arrowheads); inset shows a panoramic view at lower magnification. MAO_A_ enzyme (**D**) was found expressed in all cell layers of the seminiferous tubules (star); inset shows negative control. Scale bar in (**A**): 100 µm and (**B**–**D**): 20 µm.

**Figure 6 ijms-25-03149-f006:**
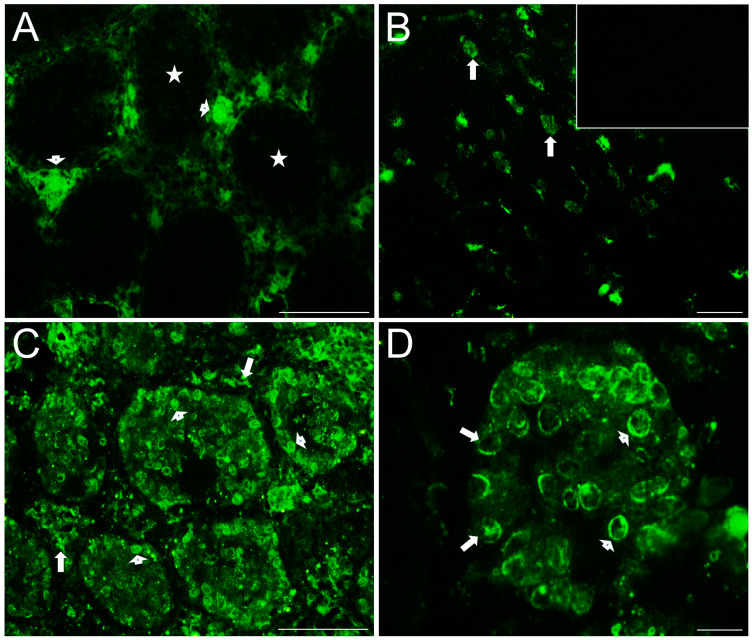
Immunofluorescence of serotonergic system elements in 150-day-old COI rabbit testes’ sections (II). Panoramic view of TPH1 enzyme immunostaining (**A**), which was found intensely depicted in clusters of Leydig cells (arrowheads) found in the base of testicular tubules (stars); positive neurons from rabbit brain stem are shown in (**B**), as a positive control, while inset shows negative control; panoramic view of 5-HT_T_ immunostaining (**C**), in which it was found intensely stained in clusters of Leydig cells (arrows) and cell membranes of gonocytes (arrowheads) and spermatocytes in the seminiferous tubules; higher magnification of a seminiferous tubule in which cell membranes of gonocytes (arrows) and spermatocytes (arrowheads) are intensely stained to 5-HT_T_ is shown in (**D**). Scale bar in (**A**,**C**): 100 µm and (**B**,**D**): 20 µm.

**Figure 7 ijms-25-03149-f007:**
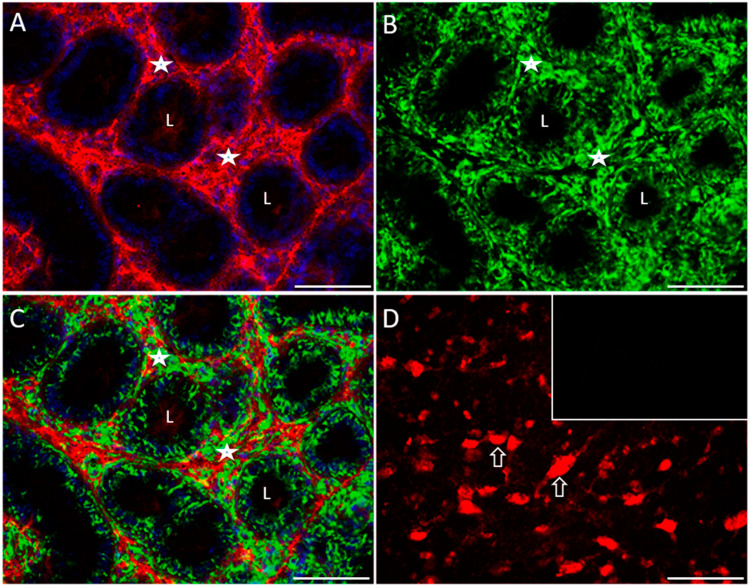
Immunofluorescence of serotonergic system elements in 150-day-old COI rabbit testes’ sections (III). Panoramic view of double immunofluorescence for 5-HT and VMAT1 transporter (**A**–**C**); 5-HT immunoreactivity (**A**) was mainly distributed along the interstitial zone (stars) and a faintly staining was found in the lumen (L) of testicular tubules,, whereas VMAT1 (**B**) was distributed in both interstitial zone and seminiferous tubules, apparently in Leydig (stars) and Sertoli cells for the evident cytoplasmic extensions, respectively; apparently, there were some Leydig cells double positive for 5-HT/VMAT1 and another only positive for 5-HT or VMAT1 as shown in the merged image (**C**); brain stem neurons (arrows) are shown as a positive control for 5-HT in (**D**). Scale bar in (**A**–**D**): 100 µm.

**Figure 8 ijms-25-03149-f008:**
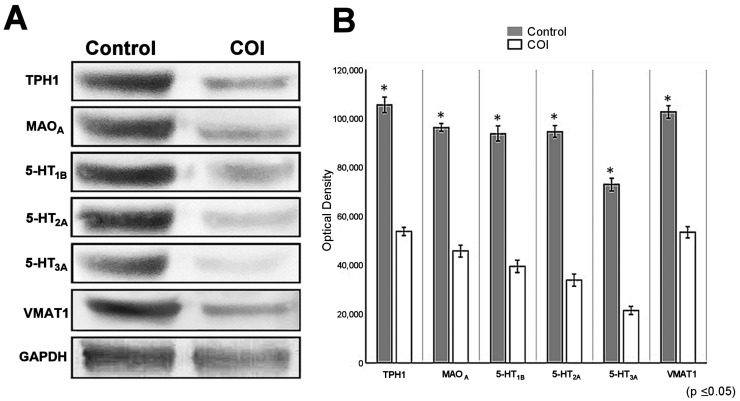
Western blotting of some serotonergic system components in control and COI rabbit testes’ homogenates. (**A**) Representative images for TPH_1_, MAO_A_, 5-HT1_B_, 5-HT2_A_, 5-HT3_A_, and VMAT1 immunoblots in control and COI rabbit testes; GAPDH was used as control; (**B**) densitometric analysis of immunoblots shown on the left. (* *p* < 0.05, Kruskal–Wallis one-way ANOVA followed by Mann–Whitney U test.)

**Figure 9 ijms-25-03149-f009:**
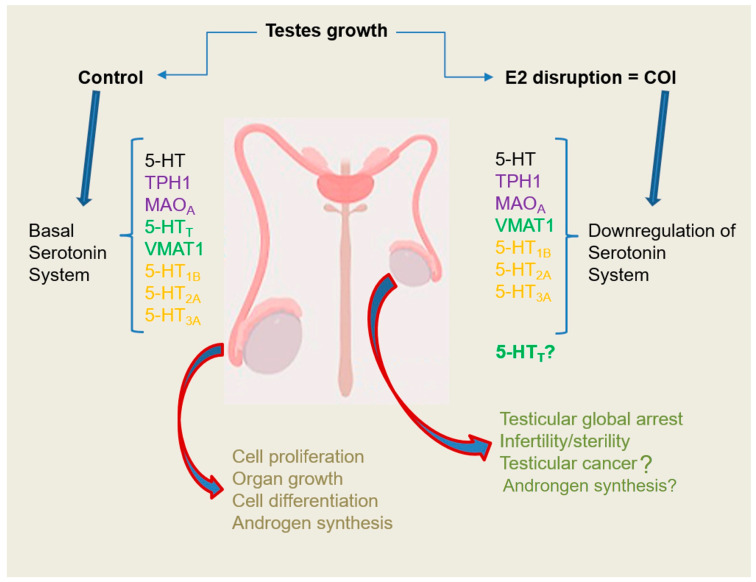
Comparative model of serotonergic system components in control and COI rabbit testes. The more relevant aspects are the finding of the presence of serotonergic system elements in control rabbits and the downregulation of them in COI rabbits.

**Table 1 ijms-25-03149-t001:** Morphometric parameters in rabbit testes with COI and Control * (*p* < 0.05).

Group	Body Weight (kg)	GSI (%)	TV (cm^3^)
Control	3.144 + 133.44	0.104 + 0.022 *	2.72 + 0.23 *
COI	2.850 + 265.54	0.011 + 0.002	0.50 + 0.21

**Table 2 ijms-25-03149-t002:** Serotonin and kynurenine pathway concentration (* *p* < 0.05).

Group	Serotonin (ng/mL)	Kynurenine (µg/mL)
Control	19.02 ± 4.78 *	123.75 ± 5.27 *
COI	13.21 ± 1.97	76.56 ± 2.93

## Data Availability

Requests for additional information should be addressed to F.J.T. and M.T.R.

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
