# Peer review of "Downregulation of Serotonergic System Components in an Experimentally Induced Cryptorchidism in Rabbits"

_ijms, 2024, doi:10.3390/ijms25063149_

Round 1

Reviewer 1 Report

Comments and Suggestions for Authors

The authors should be congratulated for their work and for addressing an important topic.

Some points warrant mention:

Major comment:

1.    Can the authors be sure that all the alterations they’ve found are due to cryptorchidism and not to the long (pre- and post-partum) period of E2 inoculation?

Minor comments:

1.    In the “Introduction” section, I suggest to avoid to report a summary of results, but to report clearly the aims of the study. Thus, the last paragraph should be rewritten.

2.    In the “Results” section, figure captions are mostly redundant with the text. I suggest to avoid findings in the figure legends and just describing what colours/asterixis/arrows/etc mean.

3.    In the “Discussion” section, line 460, “Secondly,” can be avoided.

Comments on the Quality of English Language

The English form is correct.

Author Response

Revisor 1

Major comment:

  1. Can the authors be sure that all the alterations they’ve found are due to cryptorchidism and not to the long (pre- and post-partum) period of E2 inoculation?

Yes, we can assure that the alterations generated by the inoculation of E2 are caused post-birth, for that our control group was only inoculated with corn oil (without estrogens) and they were left until they reached the age of 150 days, they always had testicles descended and without secondary alterations such as figure 1 (A-B and E-F).

Minor comments:

  1. In the “Introduction” section, I suggest to avoid to report a summary of results, but to report clearly the aims of the study. Thus, the last paragraph should be rewritten.

Thank you very much for your kind suggestion; we have done the corresponding changes to the manuscript.

  1. In the “Results” section, figure captions are mostly redundant with the text. I suggest to avoid findings in the figure legends and just describing what colours/asterixis/arrows/etc mean.

The reviewer correctly suggests that some clarification is needed in figure legends. We have reviewed the figure legends along the figures and we can change some of them, but the complexity of cell lineages present in the tissue we studied, is in need of detailed description in some figure legends, mainly in immunofluorescence images.

  1. In the “Discussion” section, line 460, “Secondly,” can be avoided.

This has been done in the latest version of the manuscript.

Note:

I take this opportunity to comment that I edited the departure of a corresponding author in the text due to a conflict of personal interest, so I can assure that at some point there will not be a conflict with the journal, likewise, I removed the name of a person in the acknowledgments section to avoid any conflicts as well.

Reviewer 2 Report

Comments and Suggestions for Authors

Interesting study. Before further processing listed comments below need to be included. 

1.„scrotum of male” – scrotum is only present in male
2.“CO exhibits changes in their expression” -what are changes in expression?- please explain
3.kynurenine pathway needs to be described in the Introduction
4. Chinchilla rabbits- the biology of this animal should be described

5. Study aim needs to be clear, not mixed with study findings. What is another species where senotonergic system was described?
5.“IDO is expressed by Principal cells, which keep the functions of the Sertoli cells but in the epididymis (80)” -It looks like Seroli cells are present in the epididymis, please rewrite
6.“The presence of some receptors and transporters that have been previously described in the reproductive system of another species suggests”-provide information on that
7. provide to the ms text information on the study limitations

Comments on the Quality of English Language

English needs attention.

Author Response

Revisor 2. Interesting study. Before further processing listed comments below need to be included. 

1.„scrotum of male” – scrotum is only present in male

This point has been changed.

2.“CO exhibits changes in their expression” -what are changes in expression?- please explain

Thank you very much for your kind suggestions; we have rewritten this section in the manuscript (see line 34).

3.kynurenine pathway needs to be described in the Introduction

Thank you very much for your kind suggestions; we have done the corresponding changes to the manuscript. (see lines 83-85).

  1. Chinchilla rabbits- the biology of this animal should be described

It is known that the sexual maturity of males of this breed of rabbit is reached at 18 weeks (4.5 months). We use the age of 150 days to ensure that they have reached sexual maturity. This point was mentioned on lines 112-114.

5.Study aim needs to be clear, not mixed with study findings. What is another species where senotonergic system was described?

Thank you very much for your kind suggestions; we have done the corresponding changes to the manuscript. In few mammalian species, an intratesticular serotoninergic system has been described; it had always been partial. In 2021, we have described a local serotonin system in adult male rats.

5.“IDO is expressed by Principal cells, which keep the functions of the Sertoli cells but in the epididymis (80)” -It looks like Seroli cells are present in the epididymis, please rewrite

We have done the corresponding changes to the manuscript.

6.“The presence of some receptors and transporters that have been previously described in the reproductive system of another species suggests”-provide information on that

This point has been clarified in lines 601-605.

  1. provide to the ms text information on the study limitations

Some limitations during our study could have been having a limited number of samples for the studies to determine the concentration of Serotonin and Kinurenine using ELISAs.

Support with other types of techniques such as in situ hybridization or FISH to establish the correlation of synthesis/presence of proteins and the DNA of their genes, this could be considered for the future.

Note:

I take this opportunity to comment that I edited the departure of a corresponding author in the text due to a conflict of personal interest, so I can assure that at some point there will not be a conflict with the journal, likewise, I removed the name of a person in the acknowledgments section to avoid any conflicts as well.

Round 2

Reviewer 1 Report

Comments and Suggestions for Authors

thank you for addressing all my comments

Reviewer 2 Report

Comments and Suggestions for Authors

I accept corrections. I think that "rabbit" should be add to keywords of the ms

Congratulations on very interesting work on synergistic system in cryptorchid rabbits.